# Title: Genetic Markers in Lung Cancer Diagnosis: A Review

**DOI:** 10.3390/ijms21134569

**Published:** 2020-06-27

**Authors:** Katarzyna Wadowska, Iwona Bil-Lula, Łukasz Trembecki, Mariola Śliwińska-Mossoń

**Affiliations:** 1Department of Medical Laboratory Diagnostics, Division of Clinical Chemistry and Laboratory Haematology, Wroclaw Medical University, 50-556 Wroclaw, Poland; katarzyna.wadowska@student.umed.wroc.pl (K.W.); iwona.bil-lula@umed.wroc.pl (I.B.-L.); 2Department of Radiation Oncology, Lower Silesian Oncology Center, 53-413 Wroclaw, Poland; lukasz.trembecki@umed.wroc.pl; 3Department of Oncology, Faculty of Medicine, Wroclaw Medical University, 53-413 Wroclaw, Poland

**Keywords:** lung cancer, carcinogenesis, genetic markers, epigenetic markers, liquid biopsy, NGS, genetic alterations, molecular landscape, microRNA, molecular heterogeneity

## Abstract

Lung cancer is the most often diagnosed cancer in the world and the most frequent cause of cancer death. The prognosis for lung cancer is relatively poor and 75% of patients are diagnosed at its advanced stage. The currently used diagnostic tools are not sensitive enough and do not enable diagnosis at the early stage of the disease. Therefore, searching for new methods of early and accurate diagnosis of lung cancer is crucial for its effective treatment. Lung cancer is the result of multistage carcinogenesis with gradually increasing genetic and epigenetic changes. Screening for the characteristic genetic markers could enable the diagnosis of lung cancer at its early stage. The aim of this review was the summarization of both the preclinical and clinical approaches in the genetic diagnostics of lung cancer. The advancement of molecular strategies and analytic platforms makes it possible to analyze the genome changes leading to cancer development—i.e., the potential biomarkers of lung cancer. In the reviewed studies, the diagnostic values of microsatellite changes, DNA hypermethylation, and *p53* and *KRAS* gene mutations, as well as microRNAs expression, have been analyzed as potential genetic markers. It seems that microRNAs and their expression profiles have the greatest diagnostic potential value in lung cancer diagnosis, but their quantification requires standardization.

## 1. Introduction

Lung cancer—i.e., bronchogenic malignant tumors stemming from airway epithelioma, is the most often diagnosed cancer in the world and the most frequent cause of cancer death. Every year, approximately 1.8 million new cases of lung cancer are diagnosed worldwide. In 2012, approximately 1.6 million people died of lung cancer and it is estimated that the number of lung cancer deaths will increase to 3 million in 2035 [1,2]. Lung cancer has a relatively poor prognosis, and the 5-year survival varies from 4% to 17%, depending on the stage of the disease at the time of its diagnosis [3]. The advancement of non-invasive diagnostics enhances the possibility of detecting lung cancer, however only 10–15% of new cases are diagnosed at its clinical early stage [4]. A total of 75% of patients are diagnosed with lung cancer at its advanced stage, when treatment options are limited. Nevertheless, in patients with clinical stage IA disease in the TNM (tumor-lymph nodes-metastasis) classification, the 5-year survival reaches approximately 60%, which indicates that a large number of patients suffer from undetectable metastases at this stage of the disease [5,6,7]. The currently used diagnostic tools—i.e., chest radiography and sputum cytology—are not sensitive enough in the diagnosis of non-small cell lung carcinoma (NSCLC), while tumor markers, such as CEA (carcinoembryonic antigen), CYFRA 21-1, NSE (neuron-specific enolase), or SCCA (squamous cell carcinoma antigen) do not make the diagnosis possible at the early stage of lung cancer [4]. These data indicate the need to find more specific, less invasive biomarkers that could be used alternatively or complementary to radiological approaches and improve lung cancer detection and the determination of its stage [8].

Lung cancer does not result from the sudden transformation of bronchia epithelioma but from the final stage of multistage carcinogenesis, with gradually increasing genetic and epigenetic changes [7,9]. The main etiological factor is the exposure to the carcinogenic components of tobacco smoke. About 90% of lung cancer cases in men and 80% in women are caused by smoking. Exposure to the xenobiotics of the tobacco smoke is associated with the modification of genetic information [10]. Nowadays, mutations that are characteristic of lung cancer and which may enable diagnosis at the early stage of the disease are sought for.

The aim of this study was to conduct an overview of the existing knowledge of the genetic markers in lung cancer diagnosis.

## 2. Genetic Markers in Diagnosis of Early-Stage Lung Cancer

### 2.1. Carcinogenesis

The better understanding of pathogenesis and the role of genetic factors in the development of lung cancer facilitates searching for morphological and molecular abnormalities characteristic not only of invasive cancer, but also for the changes referred to as preinvasive lesions in the lungs of current and former smokers without lung cancer. Morphological abnormalities, such as hyperplasia, metaplasia, dysplasia, and carcinoma in situ (CIS), may precede or accompany invasive cancer. Hyperplasia of the bronchial epithelium and squamous metaplasia have been generally considered to be reactive changes, caused by chronic inflammation and mechanical trauma. Hyperplasia and metaplasia are believed to be reversible changes which may spontaneously regress after smoking cessation. Dysplasia and carcinoma in situ are premalignant changes that frequently precede squamous cell carcinoma of the lung [7,9,11].

The molecular basis of lung cancer is the gradual accumulation of genetic and epigenetic changes in the cell nucleus. These changes lead to the weakening of the DNA structure and its greater susceptibility to subsequent mutations. Due to a neoplastic process, cells are no longer subjected to the mechanisms controlling their division and location. This is caused by irregularities in cell cycle regulation (mutations of protooncogenes and suppressor genes), and disorders in the repair processes in damaged DNA. Further changes, such as the increased expression of growth factors, sustained angiogenesis, evading apoptosis (mutations of anti-apoptotic and proapoptotic genes), limitless replicative potential, tissue invasion, and metastasis, affect tumor progression [12].

The multistage model of carcinogenesis is associated with gradually accruing molecular changes, which are shown in Figure 1 [7]. Cancer formation requires somatic alterations or “hits” that will occur only in the cancer cells [13]. The first molecular changes occurring in bronchial epithelium are microsatellite alterations. Microsatellite alterations are extensions or deletions of small repeating DNA sequences and appear as microsatellite instability (MSI, i.e., allele shift) or loss of heterozygosity (LOH), which is the absence of a normally present allele. Three or more alterations are minimally needed for a cell to transform into cancer, whereas tumor progression and metastasis lesions acquire additional DNA alterations. Molecular changes detected frequently in dysplasia are regarded as intermediate in terms of timing, and the changes detected at CIS or invasive stages are regarded as late changes. At the dysplasia stage, DNA methylation takes place [7,14,15].

Mutations are an inherent feature of lung cancer development, and their detection has a significance in both the diagnostic and treatment stages of disease. Mutations in cancerogenesis that confer growth advantage in the cancer cells are considered driver mutations. The number of driver mutations exerting an effect on the carcinogenesis is limited. Most solid tumors exhibit between 40 and 150 non-silent mutations, and most of them are regarded as passenger mutations that do not contribute to the malignant phenotype. A broad spectrum of genomic changes seen in lung cancer is associated with mutation classification, which requires a tiered approach and may enable the differentiation of driver from passenger mutations. The profound analysis of genome changes leading to cancer development enables looking for abnormalities in specific genes, which may be specific tumor markers used in diagnostics. Genomic alterations encompass mechanistic rearrangements of DNA, resulting in single nucleotide variants (SNVs) known as point mutations and small-scale deletions/insertions (indels) or copy number variants (CNVs), which reflect large-scale mutations such as amplifications, deletions, inversions, and translocations [13,16,17,18,19,20].

### 2.2. Genetic Biomarkers

Ideal reliable biomarkers should have high sensitivity and specificity, a high area under the curve (AUC) in a receiver-operator characteristic (ROC) analysis, and a high positive predictive value (PPV). A biomarker is a clinical tool for early diagnosis, prognosis, and monitoring disease evolution that enables clinical decision-making [21]. Genetic markers—i.e., changes in the structure, expression, or sequence of the genetic material—can be used to diagnose and verify the genetic predisposition to cancer and monitor the course of the disease. The development and use of DNA-based molecular markers facilitate studies on genetic variations in healthy and sick individuals. A basic attribute of genetic markers is their polymorphism—i.e., the presence of more than one allele at a specific locus (biallelic or multiallelic). The analysis of polymorphic markers makes it possible to diagnose genetically based diseases, with the unknown products of specific genes or the molecular nature of the changes leading to disease development [22,23]. The molecular markers, epigenetic markers (DNA methylation, non-coding RNA analysis), seem to be most promising because of their crucial role in the cell cycle [24].

### 2.3. Liquid Biopsy

Potential markers may be found in various biological samples—e.g., blood, urine, tissues, bronchoalveolar lavage, as well as in saliva and sputum—but none have been moved to the clinical setting yet [23,25]. A new approach in lung cancer detection is liquid biopsy—i.e., the sampling and analysis of component isolated or purified from a non-solid biological tissue. Liquid biopsy refers to the detection of tumor components in bodily fluids, such as urine, saliva, cerebrospinal fluid, and liquid cytology specimens, but primarily in blood (plasma). The blood drawing required to collect a liquid biopsy is less invasive than a tissue biopsy, what makes it easily accessible and allows the near real-time monitoring of the cancer [26]. Furthermore, tissue samples can pinpoint the exact genomic state at any tumor location, but they cannot provide a complete understanding of the tumor′s heterogeneity, while blood samples carry cell-free tumor DNA (ctDNA) from many points of tumor origin [27]. Liquid biopsy with its advantages and disadvantages is compared to conventional tissue biopsies and cytology specimens in Table 1. 

The release of ~160 base pair, nucleosome-bound fragments of DNA into circulation is a product of normal cell apoptosis. Tumor cells release their contents into the circulation too, and their amount is proportional to the overall burden of the disease. The cancer components obtained from liquid biopsy sampling are circulating tumor cells (CTCs), cell-free circulating DNA (cfDNA), and exosomes. It is also possible to isolate from blood samples such epigenetic markers as free microRNAs (miRNAs) and circulating histones and nucleosomes [8,13,24,28]. Positron emission tomography-computed tomography imaging detects tumors measuring no less than 7 to 10 mm in size and containing around 1 billion cells, while tumors containing approximately 50 million malignant cells release a sufficient amount of ctDNA to enable their detection in blood. The ctDNA released from lung cancer is detectable at levels of 0.1% to 5% of the total cfDNA. Those findings show the great potential of liquid biopsy in early stage cancer diagnosis and, hence, the selection of appropriate treatment, which explains the interest in this new technical advancement [13,29].

**Table 1 ijms-21-04569-t001:** Comparison of small histopathological and cytology specimens versus liquid biopsy.

Type of Specimen	ADVANTAGES	DISADVANTAGES	Examples of Molecular Markers	Ref.
**Small Histopathological Specimens *^,^**** **Cytology Specimens *^,^****	small biopsies and cytology samples are the basic diagnostic specimens due to the small minority of lung cancer cases that can be surgically removed (70% of lung cancers are unresectable) recent guidelines opt for the minimization of cytology use and replacing it with biopsies, which should be the gold standard in lung cancer sampling (more appropriate material for differential and molecular diagnostics)	tumor biopsies are often insufficient for molecular study or impossible to obtain small biopsy and/or cytology samples may not be representative of the total tumor because of histologic heterogeneity in the case of biopsies, multiple sampling is the requirement—minimum of 4 biopsies the rebiopsy is rarely performed, and in the view of intratumor heterogeneity a single-biosy-based analysis for personalized medicine may be a great limitation limited amount of cells in the study by Wang et al. (2015) [30], lung resection specimens had a significantly higher overall mutation rate compared to small biopsy and cytology specimens	FISH tests for *ALK* and *ROS1* can be applied to different biological specimens—biopsies or cytological samples determination of *EGFR* T790M in tumor tissue and in cfDNA are both valid alternatives; if *EGFR* T790M testing in plasma is negative, a new biopsy is recommended whenever possible; *EGFR* can be detected in plasma with a high prevalence, reflecting the landscape and heterogeneity of primary tumors and metastases	[24,30,31,32,33,34]
**Liquid biopsy**	minimally invasive tool liquid biopsy permits frequent sample collection, timely assessment of the patient′s disease status liquid biopsy offers different possible applications—response monitoring, tumor recurrence detection, determination of residual disease after full tumor resection, early detection of lung cancer, and immuno-oncology liquid biopsy enables testing for genetic and epigenetic abnormalities specific to the tumor, and provides abilities to identify mutations in both primary and metastatic lesions—liquid biopsy represents the whole genomic picture of the tumor a new source for cancer biomarkers	validation and clinical usefulness are not sufficiently determined as yet lack of standardization detection techniques require a high sensitivity in order to detect the DNA from tumor cells negative results require testing with conventional techniques, such as tumor biopsy hemolysis may influence the results	microRNAs are most commonly assessed in patient′s serum or plasma; an example is a panel of circulating microRNAs in the study by Sromek et al. (2017) [35]—the combination of miR-9, miR-16, miR-205, and miR-486 can serve as potential NSCLC biomarkers with 80% sensitivity and 95% specificity	[24,35,36]

* direct comparisons between small histopathological specimens and cytology samples are limited, and both of these specimens appear to perform similarly, with the high feasibility of molecular testing; ** whenever possible, cytology should be interpreted in conjunction with the histology of small biopsies, as the 2 modalities are complementary, in order to achieve the most specific and concordant diagnosis; *ALK* - anaplastic lymphoma kinase; cfDNA - cell-free circulating DNA; *EGFR* - epidermal growth factor receptor; FISH - fluorescence in situ hybridization; NSCLC - non-small cell lung carcinoma; *ROS1* - c-ros oncogene 1.

A retrospective study reported by East Carolina University researchers compared standard molecular analysis strategies to liquid biopsy. The analysis showed that liquid biopsy can provide results within 72 h, enabling lung cancer patients to start targeted therapy within a median of eight days from diagnosis [37].

## 3. Advancement of Molecular Strategies and Techniques Used to Identify Lung Cancer Genetic Markers

In recent years, the advancement of molecular strategies and analytic platforms, including genomics, epigenomics, and proteomics, has been observed, enabling the analysis of the genome changes which play a key role in the pathogenesis and progression of lung cancer. Cancer genomic research unlocks possibilities in the understanding of somatic modifications in cancer that may be used as a tool in prevention, early diagnosis, novel treatments, and resistance monitoring. Hence, genomic testing can help to identify genomic changes as potential biomarkers of lung cancer. However, the broad spectrum of genomic alterations and lack of a universal technique for their detection requires a tiered approach to their analysis (Table 2) [17,23,25,38].

### 3.1. Genomics

In 1982, the first human, naturally occurring tumorigenic somatic mutation was discovered. That discovery and the advancement of molecular technologies gave the beginning of genomics, leading to the completion of the first human reference genome and the first human cancer genome by the Human Genome Organization (HUGO) [39]. Genomics deals with organisms’ genome analysis, and its aim is to determine the DNA sequence, map the genome, and determine the dependencies and interactions within the genome. Genomics analyzes genetic material in a multi-complex way by testing many genes at the same time [23,40]. The discovery of the duplex DNA structure and its complementary rules in 1953 gave the beginning for the development of new molecular diagnostic technologies based on four major techniques [41]:(a)Enzymatic DNA restriction—there are four classes of restriction enzymes. The most commonly exploited are the enzymes belonging to class II, which require only ions of Mg2+ to recognize their target DNA sequence and cleave it. The usage of different combinations of restriction enzymes allows us to characterize and manipulate DNA in fundamental DNA technology approaches such as cloning or mapping [42,43].(b)Nucleic acid hybridization—in situ hybridization (ISH) uses labeled nucleic acid probes to detect specific DNA or RNA targets in tissue sections, intact cells, or chromosomes. The basic principle underlying ISH is the ability of single-stranded DNA or RNA to anneal specifically to a complementary sequence and form a double-stranded hybrid. Nucleic acid hybridization is the foundation of Southern or Northern blot hybridization and microarray technology [44,45,46]. The development of microarray technology (also known as DNA microarrays, DNA chips) is connected with the transition of molecular biology into postgenomic era by enabling large-scale genotyping and gene expression profiling [47,48].(c)Polymerase chain reaction (PCR)—PCR allows for the exponential amplification of specific targeted genetic loci in a reaction mixture containing DNA primers, deoxynucleotides (dNTPs), and DNA polymerases. PCR is a qualitative technique to amplify and copy a targeted area of extracted DNA a million to a billion-fold over [49]. In the course of time, modifications and advances in this molecular diagnostics technique enabled the relative or even absolute quantification of DNA through the usage of quantitative real-time PCR (qPCR) or partition-based PCR techniques, such as droplet digital PCR (ddPCR) [27,49]. The exploitation of PCR in quantitative DNA analysis is leading to the increased clinical usefulness of PCR for a broad range of applications; PCR is used in a variety of methods, such as allele-specific PCR-based methods or mutation screening methods, including melting curve analysis, that are used in the analysis of mutations sequences. PCR is also used in pyrosequencing and next-generation sequencing (NGS) as a pre-step that provides the sequencing of the generated PCR products [17],(d)Fluorescence-based methods—the use of hybridizing fluorescent-labeled probes is one of the advancements in cytogenetics. Fluorescence-based methods include the fluorescence in situ hybridization (FISH) and comparative genomic hybridization (CGH) techniques, which detect large-scale amplifications and deletions. FISH uses specific fluorescent probes that bind to nucleic acid sequences with a high degree of sequence complementarity, helping one to localize these DNA sequences on chromosomes. Microarray-based CGH (array CGH) enables us genome-wide screening for chromosomal imbalances on the basis of genomic DNA hybridization to complement probes that are immobilized on a slide [17,23,49].

### 3.2. Understanding of Molecular Pathology of Lung Cancer

The analysis of the lung cancer genome is focused on mutation detection. The identification of the cancer gene targets of each of somatic copy number alterations (SCNAs) is an important challenge. The majority of tumors still lack of an identified molecular driver that could be used as genetic biomarkers. Comprehensive testing looking for a large number of genomic alterations may provide clinically significant information that will allow us to impact the therapeutic options and patients’ prognosis. Because SCNAs may overlap and target different genes in different cancer types, all SCNA candidates should be tested in an appropriate model system [39,50].

Past techniques that have been successfully used in the field of the cancer genome exploration are capillary electrophoresis-based “Sanger” sequencing, the array-based genome-wide analysis of amplifications and deletions, gene expression arrays, and retrovirus-mediated expression screening techniques. Until recently, the vast majority of techniques used widely for mutation detection were single analyte assays. Single analyte tests are still indispensable; however, the introduction of high-throughput NGS, also referred to as massively parallel sequencing, quickly captured the attention of molecular diagnosticians in the past decade. While single analyte assays are the combination of the amplification and analysis of a target of interest (DNA, RNA) in a single scope, a high-throughput NGS allows multiplex PCR with the simultaneous amplification of a pre-specified panel of genes in a single reaction [17,51,52]. There is a large number of potential applications for NGS technologies and different NGS tests depending on the application purpose. NGS enables the comprehensive genome or exome examination of any cancer through the parallel sequencing of millions of short reads of any type of nucleic acid (including micro-RNA and other non-protein coding DNA species). The scope of NGS can range from whole-genome sequencing (WGS) and whole-exome sequencing (WES), through RNA expression profiling, to targeted oncology panels covering either several or hundreds of genes:WGS and WES to find novel mutations in so far unreported gene loci;paired-end, mate-pair sequencing to identify structural variations;targeted sequencing for mutation discovery and validation;transcriptome sequencing for the quantification of gene expression and discovery of transcribed mutations;small RNA-sequencing to microRNA profiling;large scale analysis of DNA methylation and immunoprecipitation for the genomic mapping of DNA–protein interactions [39,53,54].

The NGS of WGS or WES is used for research purposes, while the sequencing of targeted gene panels is very common in clinical practice for the purpose of finding targetable genomic alterations. In clinical practice, NGS may enable the detection of multiple targets or alteration types, such as mutations, gene copy changes, and rearrangements [18]. In Table 2 is presented a list of techniques used in genomic analysis on the basis of mutation variants.

The molecular diagnostics of lung cancer, similarly to NGS applications purposes, is explored in two areas—for research purposes, to find novel druggable mutations, and in clinical practice to diagnose and select eligible patients for specific tyrosine kinase inhibitor (TKI) therapy [18,55]. Before the development of NGS, our understanding of the molecular pathology of lung cancer was based on such techniques as mismatch repair detection, the sequencing of candidate genes, single nucleotide polymorphism (SNP) arrays, and gene expression analysis. The availability of NGS has enabled the full mutation characterization of lung cancer in the International Cancer Genome Consortium and the Cancer Genome Atlas projects. These projects are engaged in the development of personalized medicine and the availability of targeted therapies for patients with adenocarcinoma (AC) and squamous cell carcinoma (SCC) in the first instance. The sequencing of the lung cancer genome may identify unknown variants along with the known mutations. Furthermore, it may allow the detection of less common oncogenic alterations with available targeted therapies. The National Comprehensive Cancer Network Guidelines recommends testing a panel of genes for NSCLC, which consists of epidermal growth factor (*EGFR*) mutations, anapestic lymphoma kinase (*ALK*) rearrangements, and c-ros oncogene 1 (*ROS1*) rearrangements. These biomarkers are considered the “must-tests” biomarkers in lung cancer patient diagnosis and are analyzed by single-gene assays such as PCR, immunohistochemistry (IHC), and FISH [20,55,56,57,58,59,60]. Sanger sequencing, qPCR, ddPCR, IHC, and FISH are regarded as the gold standard techniques of molecular analysis in clinical practice, while tumor-only sequencing, matched-tumor, and normal-tissue sequencing are the gold standards in somatic mutation detection [18,61].

In the study by Park et al. (2020) [55], the researchers have compared single-gene assays, such as real-time PCR, IHC, and FISH (considered as the gold standard for selecting eligible patients for *EGFR*-, *ALK*-, and *ROS1*-specific TKI therapy) to targeted NGS. Considering the NGS results as the final outcome, *EGFR* PCR revealed a sensitivity of 80.3% and specificity of 99.4%, *ALK* FISH showed a 71.4% sensitivity and 100% specificity, and *ROS1* FISH showed a 100% sensitivity and 99.5% specificity. These results are related to the lower sensitivity of single-gene assays in comparison to deep-targeted NGS. The data also revealed the necessity of revalidating the results of the single-gene assays, especially for negative *EGFR* assays.

### 3.3. Epigenomics

Epigenomics is the study of the complete set of epigenetic modifications on the cell genetic material. Epigenetic mechanisms play a critical role in the regulation of gene expression and are prompted by an environmental exposure, such as an exposure to the carcinogenic components of tobacco smoke [23,40,62]. Chromatin is composed of DNA, histone proteins, and other nuclear proteins. Under the influence of external factors, chromatin undergoes a number of reversible covalent modifications. The chromatin modification mainly comprises histone posttranslational modifications (PTMs) and DNA methylation. These chemical changes play an important role in the gene expression and regulation, which are involved in cellular processes such as differentiation and maturation. Methylation modifies the histone by strengthening the charge of the spool, which leads to the more tight packing of the DNA around the spool and makes the DNA less accessible from being read—“turning off” the gene expression [63]. Epigenomics is currently a field of intense research and it fosters development in molecular cancer therapies. The methods of analysis depend on the nature of the biomarker and the availability of new technologies that are able to perform high-throughput experiments, such as microarrays and NGS in the measurements of DNA methylation, miRNAs, or HPLC coupled with mass spectrometry in the detection of histone PTMs [21].

DNA strands undergo various chemical reactions. DNA modifications, such as 5-methylcytosine (5mC), 5-hydroxymethylcytosine (5hmC), 5-formylcytosine (5fC), and 5-carboxylcytosine (5caC) have been applied to analytical techniques such as mass spectrometry, molecular imaging, pulldown assay, and DNA sequencing. DNA methylation is the addition of the methyl group (-CH_3_) at the 5th carbon position of cytosine bases, which are located 5′ to a guanosine in a CpG dinucleotide. The DNA methylation of cytosine (5mC) can be mapped genome-wide with the use of several methods, such as methylation-specific restriction enzymes or affinity purification, or by using bisulfite conversion followed by sequencing (BS-Seq). The methylation-specific FISH (MeFISH) was the first visualization of 5mC. The hydroxymethylation of cytosine (5hmC) is also thought to have a role in epigenetic gene expression and has been analyzed with the usage of modified bisulfite sequencing methods, 5hmC-specific restriction enzymes, or immunoprecipitation [24,63,64,65,66]. Alternative techniques in the analysis of chemically modified DNA strands are multiplex ligation-dependent probe amplification (MLPA) assays that are based on the bisulfite conversion of DNA and qPCR-based reactions, such as methylation-sensitive high-resolution melting (MS-HRM) or pyrosequencing [21]. The analyses of histone variants and PTMs are a combination of molecular biology techniques that involve western blotting, ELISA analysis, mass spectrometry, and histone modification assays in the main. These combinations are the coupling of chromatin immunoprecipitation (ChIP) technology with DNA microarrays (ChIP-Chip) or with NGS (ChIP-Seq). The standard way to detect circulating histones is a direct measurement of histones by the use of immunoassays in plasma samples [8,63,64].

## 4. Genomic and Epigenomic Changes in Lung Cancer Diagnosis

Oncogenic transformation is a highly complex, multi-step process involving genomic and epigenomic alterations. Lung cancer is characterized by a high tumor mutational burden (TMB) in comparison with other cancer types, which is probably related to smoking habits and an exposure to the xenobiotics of tobacco smoke [12,67]. Kan et al. (2010) [68] studied 441 tumors by paying special attention to the rates of protein-altering mutations. Among the tumor types analyzed, they found that NSCLC is one of the tumors with the highest rate of protein-altering mutations, with rates in adenocarcinomas and squamous cell carcinomas (SCCs) of 3.5 and 3.9 per megabase (Mb), respectively. On the other side, prostate cancer was characterized by a low mutation rate of 0.33 per Mb in comparison with an average rate of 1.8 per Mb of DNA across all tumor types.

Studies that are concerned with mathematical modeling related to the clonal mutation burden in several cancer types showed that lung cancer reflects predominantly mutations accumulated at the early stages of tumorigenesis. Lung tumors are characterized by extensive genomic aberrations, including aneusomy, the gains and losses of large chromosome regions, gene rearrangements, copy number gain, or amplifications [67].

### 4.1. Oncogenes and Tumor Suppressor Genes in Lung Cancer

There are two classes of cancer genes: oncogenes and tumor suppressor genes. Most oncogenes began as proto-oncogenes—i.e., regulatory proteins involved in cell growth and proliferation or the inhibition of apoptosis. The activation of oncogenes leads to uncontrolled cellular proliferation, and cells undergo oncogenic transformation. The most frequently mutated proto-oncogenes in lung cancer are those from the *MYC*, *RAS*, and *HER* families. Tumor suppressor genes or antioncogenes are a group of genes controlling cell growth through inhibiting cellular proliferation and maintaining genome stability. In lung cancer, the most frequent alterations among tumor suppressor genes are the mutations of *TP53*, *RB*, and *p16* [14,15,69]. Ahrendt et al. (1999) [15] used four PCR-based techniques (p53 sequencing, K-ras mutation ligation assays, K-ras-enriched PCR, p16 methylation-specific PCR, and microsatellite analysis) to examine the presence of cancer cells in bronchoalveolar lavage (BAL) fluid. They compared the results of genetic alterations obtained from different kinds of samples—tumor tissue, blood, and BAL fluid. *p53*, *KRAS* (Kirsten rat sarcoma viral oncogene) mutations, microsatellite instability, and *p16* methylation were found in 56%, 27%, 46%, and 38% of patients with NSCLC, respectively. Their results also showed a higher frequency of *p53* mutations in SCC than in adenocarcinoma, and a higher frequency of *KRAS* mutations in patients with the adenocarcinoma subtype.

In the early days of understanding lung cancer genetic aberrations, most of the discoveries were confined exclusively to adenocarcinoma, and only recently have NGS technologies allowed the better molecular characterization of other histotypes. The first recognized mutations in NSCLC were identified in *KRAS* and *TP53*. Then, in 2004 mutations in the kinase domain of *EGFR* were described that changed the lung cancer treatment paradigm. Both *KRAS* and *EGFR* mutations are identified almost exclusively in lung adenocarcinomas, similarly to rearrangements involving *ALK* and *ROS1*, which were described in 2007. *KRAS* was not only the first described mutation but also the most frequently mutated oncogene in NSCLC (20–35%). Activating *EGFR* mutations are found in about 15% of NSCLC, while *ALK* and *ROS1* alterations are quite rare ( < 5% of lung cancers) but are frequent among light to never smokers. These mutations—*EGFR*, *ALK*, and *ROS*—may predict a good response to treatment with the specific tyrosine kinase inhibitor (TKI). The number of clinically relevant genomic alterations with already available or newly developed kinase inhibitors is rapidly increasing. There are a number of other important recognized oncogenic alterations in lung adenocarcinoma that may be also used as potentially targetable alterations, including B-Raf proto-oncogene (*BRAF*), Erb-B2 Receptor Tyrosine Kinase 2 (*ERBB2*), mesenchymal-epithelial transition factor (*MET*), and rearranged during transfection (*RET*). Current technologies such as the NGS approach reveal multiple gene alterations in a single tumor, however the co-occurrence of more than one oncogenic driver is infrequent [20,26,56,57,58,59,60]. The data collected in the study of the Lung Cancer Mutation Consortium (Sholl et al., 2015) [70] revealed that, among 1007 lung adenocarcinomas, the co-occurrence of more than one oncogenic driver was found in only 3% of patients. Originally, the molecular diagnostics of squamous cell carcinoma (SCC) was based on the absence of certain targetable genomic alterations that are commonly found in adenocarcinoma—i.e., *EGFR* and *ALK* [59]. Devarakonda et al. (2018) [71] analyzed the molecular profiles, consisting of 1538 genes, of 908 patients with NSCLC. Their results revealed differences in mutation frequency between adenocarcinoma and SCC, and as was expected, the characteristic activating mutations of adenocarcinoma such as *KRAS*, *HRAS*, *NRAS*, and *EGFR* were identified only in 3% of SCCs. The commencement of NGS technology also provided an opportunity to better understand the mutational landscape of specific genomic alterations in SCC diagnosis. Current studies led to the identification of numerous recurrent changes in SCC, such as gene amplifications (*CCND 1-3, CDK4, FGFR 1-3, MET, PDGFRA, PIK3CA, SOX2*), gene fusions (*FGFR3-TACC3*), tumor suppressor mutations (*PTEN, TP53*), and point mutations (*EPHA2, AKT1, DDR2*), often in combination and some of them with available therapies (Table 2) [59,68]. To sum up, in lung cancer the current National Comprehensive Cancer Network (NCCN), Domestic Lung Cancer Clinical Guidelines, and National Health and Family Planning Commission Diagnosis and Treatment Norms suggest that some driver gene variants, including the mutation of *EGFR*, *KRAS*, *BRAF*; the mutation or amplification of *HER2* (human epidermal growth factor receptor 2), *ALK*, *ROS1*, and *RET* rearrangements; and the *MET* copy number amplification or variable shear variations in *MET* exon 14 are the essential parts of the “core gene list” [20]. In Table 3, we list the molecular landscape of lung cancer with the currently available targeted therapies.

### 4.2. Microsatellite Markers

Another aspect of lung cancer molecular diagnosis is the analysis of microsatellite markers, potentially used to assess clonality. Miscrosatellite instability (MSI) is an effect of defective DNA mismatch repair (MMR) and has been implicated in tumorigenesis. MSI was initially noted in the colon cancers of patients with hereditary nonpolyposis colon cancer (HNPCC). The most common mutations identified in HNPCC involve MLH1, MLH3, MSH2, MSH3, MSH6, PMS2, and POLE [83,84]. At the beginning of understanding the lung cancer molecular landscape, Sozzi and et. (1999) [85] investigated the frequency and extent of MSI and LOH in the plasma DNA of NSCLC patients with limited disease. The study focused on four markers detecting shifts or LOH—tetranucleotide repeat (D21S1245), recognized as being prone to microsatellite instability in various cancer types and dinucleotides D3S1234, D3S1300, and D3S4103 located in the introns of the *FHIT* (fragile histidine triad diadenosine triphosphatase) gene. *FHIT* is a tumor suppressor protein which participates in apoptosis induction (mediated by the activation of death receptors—DRs—and the caspase cascade signaling pathways) and the regulation of the cellular cycle. The overexpression of *FHIT* in NSCLC cells leads to the higher expression of DRs and the activation of caspase 3, caspase 8, and caspase 9. The *FHIT* gene encompasses the common fragile site FRA3B on chromosome 3 and shows a high rate of LOH in lung cancer, particularly in smokers [86,87]. It was found that 56% (49 of 87) of NSCLC tumors showed microsatellite alterations (shift or LOH) in at least one locus. MSI was recognized as the appearance of new tumoric allele(s) which are not present in normal DNA. LOH was scored if the allele signal was reduced to less than 30% of its normal intensity. A total of 30 out of 49 (61%) patients with microsatellite alterations in tumor DNA also showed microsatellite changes in their plasma DNA [85].

### 4.3. Epigenetic Changes in Lung Cancer

An enormous field of lung cancer molecular landscape is determined by epigenetic changes. The last two decades have seen exponential developments in the epigenetic understanding of lung cancer. The two main groups of epigenetic modifications are DNA methylation (hypermethylation or hypomethylation) and non-coding RNA expression, known as microRNAs (miRNAs). These epigenetic disruptions may represent reliable biomarkers for lung cancer risk assessment, early diagnosis, prognosis stratification, molecular classification, and the prediction of treatment efficacy. Aberrant DNA methylation is catalyzed by three DNA methyltransferaze enzymes (DNMTs) and promotes carcinogenesis through the promoter methylation of tumor suppressor genes, leading to silencing their expression. DNMT1 expression is increased in early stage lung cancer and is implicated in its pathogenesis through the silencing of such genes as *RASSF1A* and *CDKN2A*. *CDKN2A* was not only the first tumor suppressor gene to be found inactivated in lung cancer, but it shows promoter methylation in almost all cancers [24,65,88]. Moreover, *CDKN2A* is prone to hypermethylation early during lung cancer development, for which Palmisano et al. (2000) [89] provided scientific evidence. In their study, *CDKN2A* methylation was evident in two sputum samples which had been collected from subjects almost three years prior to diagnosis [65]. In lung cancer, the methylation status of over 40 genes has been assessed in tumors, cell lines, sera, and/or sputum. In more recent works, there was the demonstration of an increased frequency and level of promoter hypermethylation in a number of genes involved in cell cycle regulation, adhesion, apoptosis, and signal transduction. The lung cancer epigenetic diagnostics may be based on the assessment of typical hypermethylated genes or the assessment of the hypermethylation of genes that are methylated in other cancers than lung cancer. In lung cancer, there is a little or no methylation of genes commonly methylated in other cancers, such as *ARF*, *CDKN2B*, *CTTNB1*, *MLH1*, and *RB1*. The hypermethylated genes in lung tumors such as *APC*, *CKDN2A* (encodes p16INK4A and p14arf), *CHD13*, *RARB*, and *RASSF1A* may be considered as tumor suppressors [65,88]. Ooki et al. (2017) [90] evaluated a panel of cancer-specific methylated genes in tumors and adjacent normal lung tissue from NSCLC patients. A panel consisted of six genes—*SOX17, HOXA9, AJAP1, PTGDR, UNCX*, and *MARCH11*—and showed a high sensitivity (96.7%) and specificity (60%) [24]. Another panel of possible genes associated with NSCLC was established by Liu et al. (2018) [91]. The combination of *PCDHGB6, HOXA9, MGMT*, and miR-126 was characterized by the highest sensitivity (85.2%) and specificity (81.5%) among such genes as *PCDHGB6, HOXA9, MGMT,* miR-126, *SOCS2*, and *NORE1A* [24]. A meta-analysis performed by Huang et al. (2016) [92] focused on generating a list of differentially methylated genes among NSCLC histotypes. Their results showed two hypomethylated genes (*CDKN2A* and *MGMT*) and three hypermethylated genes (*CDH13, RUNX3, APC*) in adenocarcinomas compared with SCC, with the higher sensitivity and specificity values of *CDH13* and *APC* [24].

### 4.4. MicroRNAs in Lung Cancer Diagnosis

The rapid development of genomics and epigenomics is associated with the increasing popularity of microRNA. Basic techniques used to detect and investigate the expression of miRNAs are northern blotting, RT-PCR (reverse transcriptase PCR), and microarrays [23,93]. RT-qPCR using TaqMan miRNA assays is the gold standard in miRNA quantification [94]. miRNA is not tissue-specific. However, during oncogenesis, tumor cells develop a unique genetic profile. The profiles of circulating miRNAs are different for each cancer microenvironment and tumor progression stage. The analysis of these profiles enables the better understanding of tumor etiopathogenesis and the origin of cancer. The profiles of miRNAs expression are useful in the case of poorly differentiated cancers and metastases with unknown primary origin. Some studies revealed that circulating miRNAs also play an important role in the treatment and prognosis of lung cancer [93,94]. microRNAs are a class of small, single-stranded, non-coding RNAs that post-transcriptionally regulate the translation of target genes and influence a series of cellular functions, such as proliferation, differentiation, and apoptosis; therefore, an altered expression of miRNAs in different cancer types can affect the deregulation of cellular activities. microRNAs develop a specific gene expression profile for individual tissues. microRNAs represent ideal markers which can be evaluated in biological fluids because of their high stability. microRNAs’ resistance to endogenous and exogenous RNase activity, extreme temperatures, repeated freeze-thaw cycles, and pH changes enable their effective isolation in biological fluids and measurement using RT-qPCR. microRNAs’ participation in both physiological (proliferation, differentiation, apoptosis) and pathological processes (inflammation, cancers) plays an important role in oncogenesis. microRNAs may also constitute useful diagnostic and prognostic markers for cancer diagnosis and treatment, as well as serve as potential therapeutic targets or tools. They may act as oncogenes or tumor suppressor genes, and are defined as oncomicroRNA and anti-oncomicroRNA. microRNAs’ expression profiles in lung cancer are different in comparison with those in healthy tissues and with those in other types of cancer. This makes it possible to consider creating microRNA expression profiles that will be specific for lung cancer and will confirm the disease [93,94,95,96]. In one of these studies, Xing et al. (2010) [97] developed a panel of microRNAs that could be used as a sputum-based test for the early stage SCC of the lungs. During the study, they identified three microRNAs (miR-205, -210, and -708) that were overexpressed and three microRNAs (miR-126, -139, and -429) that were underexpressed in tumor specimens. miR-205 was the best single microRNA, resulting in a 65% sensitivity and 90% specificity, while the combination of three microRNAs—miR-205, -210, and -708—provided the best prediction and revealed a sensitivity of 73%. Yu et al. (2010) [98] also developed a panel of microRNAs, but this one could be used as highly sensitive and specific sputum markers for the early detection of lung adenocarcinoma. The researchers found three microRNAs (miR-486, miR-126, and miR-145) that were underexpressed and four microRNAs (miR-21, miR-182, miR-375, and miR-200b) that were overexpressed in tumor specimens. The combination of four microRNAs—miR-486, -21, -200b, and -375—provided the best prediction and resulted in a significantly higher sensitivity and specificity, at the level of 80.6% and 91.7%, than the sensitivity and specificity of the individual microRNAs. Another study that focused on the identification and construction of microRNA panels for lung cancer diagnosis was the study by Lu et al. (2018) [5]. The researchers constructed two panels consisting of the following microRNAs: model A—miR-17, miR-190b, miR-19a, miR-19b, miR-26b, and miR-375; and model B—miR-17, miR-190b, and miR-375. Model A was built in order to evaluate the risk of lung cancer and was characterized by a high sensitivity (81%) and specificity (80%). All six microRNAs constituted effective predictors of lung cancer, with higher expression levels in the lung cancer group than in the asymptomatic high-risk group. Model B was built in order to estimate the probability of SCLC, with a sensitivity of 81% and a specificity of 80%. Zheng et al. (2011) [99] investigated the potential of circulating plasma microRNAs for the early detection of lung cancer. Out of the 15 selected microRNAs, 6 microRNAs (miR-155, miR-197, miR-182, miR-21, miR-128, and miR-183) were significantly elevated in the plasma of patients with lung cancer and subjected to further analysis. The combination of three microRNAs—miR-155, miR-197, and miR-182 showed an 81.33% sensitivity and 86.76% specificity. This panel was also measured in an independent set of 14 patients with lung cancer during the early and late phase of chemotherapy in order to explore its value in the monitoring of the treatment effectiveness of chemotherapy. The levels of miRNA-155, miR-197, and miR-182 were significantly reduced after treatment in the majority of responsive patients. In the study by Xi et al. (2018) [4], the researchers investigated the differences in the expression levels of 12 microRNAs between benign pulmonary nodules and malignant pulmonary nodules. The expression levels of miR-17, miR-146a, miR-200b, miR-182, miR-221, miR-205, miR-7, miR-21, miR-145, and miR-210 were significantly higher in the NSCLC patients, with a sensitivity at the level of 54.8–83.3% and a specificity in the range of 60.0–86.7%. In other study, Sromek et al. (2017) [35] found that combined miR-9, miR-16, miR-205, and miR-486 can serve as potential NSCLC biomarkers with an 80% sensitivity and 95% specificity. In the study by Hennessey et al. (2012) [100], the feasibility of using serum microRNAs as non-invasive biomarker assays in the early detection of lung cancer was examined. The results yielded five candidate microRNA pairs that were significantly differentially expressed between the NSCLC and healthy controls, with a sensitivity and specificity of at least 75%. A combination of two differentially expressed microRNAs—miR-15b and miR-27b—was able to discriminate NSCLC from healthy controls with a 100% sensitivity and 84% specificity. Heegaard et al. (2012) [101] focused on the measurement of 30 different circulating microRNAs that had been previously reported to be differently expressed in lung cancer tissue. The researchers found seven microRNAs which had statistically significant lower expressions in lung cancer patients (miR-146b, -221, -let7a, -155, -17-5p, -27a, and -106a) and one which was significantly increased (miR-29c). Despite significant differences in the microRNA expression levels between the study group and the control, the expression profiles could not distinguish the study group from the control group accurately. The accuracy of the best predictive panel of microRNAs was only 57–60%. Ulivi et al. (2019) [95] focused on early stage (IA-IIIA in the TNM classification) NSCLC to develop a microRNA panel as a potential prognostic biomarker in patients undergoing surgery. In a univariate analysis, five microRNAs—miR-26a-5p, miR-126-3p, miR-130b-3p, miR-205-5p, and miR-21-5p—were significantly associated with disease-free survival (DFS) in SCC patients, and four of these microRNAs (miR-26a-5p, miR-126-3p, miR-130b-3p, and miR-205-5p) were also associated with overall survival (OS). In adenocarcinoma patients, only miR-222-3p, miR-22-3p, and miR-93-5p were significantly associated with DFS, and miR-196-3p was associated with OS. The study showed that miR-126-3p played an independent prognostic role associated with a lower risk of relapse or death due to SCC. Previous reports have demonstrated that miR-126 may function as an important regulatory factor in the development of NSCLC. Xu et al. (2017) [102] also evaluated the predictive value of microRNAs in terms of the DFS and OS of patients with NSCLC. The researchers focused on angiogenic microRNAs, such as miR-18a, miR-19a, miR-20a, miR-92a, miR-126, miR-130a, miR-210, miR-296, and miR-378, that play an important role in tumorigenesis and angiogenesis. The comparison between microRNAs in the patients with NSCLC and the healthy controls showed that the plasma miR-18a and miR-126 expression levels were lower in the patients with NSCLC, whereas the expression levels of miR-19a, miR-20a, miR-92a, miR-130a, miR-210, miR-296, and miR-378 were higher in the patients with NSCLC. The low plasma expression levels of miR-18a, miR-20a, miR-92a, and miR-126 were correlated with a prolonged DFS, whereas high plasma miR-18a, miR-20a, miR-92a, miR-210, and miR-126 expression levels were correlated with a shorter OS. In another study, Yan et al. (2019) [103] investigated the expression and clinical significance of miRNA-99a and miRNA-224 in the serum of patients with NSCLC. The expression level of miR-99a was remarkably lower in the NSCLC patients than in the control group, and it was significantly correlated with the pathological stage, the presence and absence of lymph node metastasis, and the tissue differentiation. The expression of miR-224 was significantly higher in the NSCLC patients in comparison to the healthy individuals, and it was also correlated with the pathological stage, the presence and absence of lymph node metastasis, and the pathological grade. Aiso et al. (2018) [94] analyzed microRNA expression before and after surgical resections of NSCLC. In the first phase of the study, the researchers evaluated the expression of miR-145, -20a, -21, and -223, which were previously reported as candidate diagnostic markers of NSCLC. In this phase, they revealed a significant reduction in the miR-145 and miR-20a sera levels in patients with NSCLC (of all groups—i.e., with stage I-II, III, or IV NSCLC) in comparison to the control group. The miR-21 differences with significant increases were observed only between the patients with stage-IV NSCLC, in comparison with the patients suffering from NSCLC of stage I-II. The miR-223 was remarkably higher in patients with stage-IV NSCLC in comparison with the control group. In the second phase of the study, the expression levels of miR-145, -20a, -21, and -223 were evaluated after tumor resection. The expression of miR-145 and miR-20a was significantly increased after resection in comparison with the pre-resection levels, and it was similar to the levels of microRNAs in the control group. An ROC (receiver operating characteristic) curve analysis revealed that miR-145 was the most suitable diagnostic marker for NSCLC that distinguishes the NSCLC patients of all stages from the healthy individuals with a high sensitivity and specificity. Szczyrek et al. (2019) [104] evaluated the diagnostic value of selected plasma microRNA (miR-27a-3p, miR-31, miR-182, miR-195) expression complementary to Drosha and Dicer in lung cancer patients. Dicer and Drosha enzymes play an essential role in microRNA biogenesis and the conversion of pri-miRNA into pre-miRNA. The expressions of miR-27a-3p, miR-21, and miR-182 were significantly higher in the study group than in the healthy volunteers, whereas the miR-195 expression was significantly lower in the lung cancer group. The increased expression of miR-27a-3p (89% sensitivity and 77% specificity), miR-31 (73% sensitivity and 61% specificity), and miR-182 (70% sensitivity and 79% specificity) reported in the study could contribute to a reduction in the activity of the Dicer and Drosha enzymes and to a reduction in the expression of microRNAs described as tumor suppressor genes.

## 5. Lung Cancer Genetic Heterogeneity

The use of molecular technology and analysis of the cancer genome showed that solid tumors are genetically heterogeneous among individuals with the same tumor type (interpatient heterogeneity); there is observed a diversity of tumor cells within a single tumor (intratumor heterogeneity) and diversity between the primary tumor and its metastasis (intertumor heterogeneity) [68]. This means that cancer subtypes may contain different molecular variants and that a wide range of genomic variants may exist within a single tumor subtype [19,27]. Mutations of *KRAS* and *TP53* were the earliest recognized mutations in NSCLC. In general, colon, pancreas, and lung carcinoma are associated with *KRAS* mutation, while somatic *TP53* mutations occur in almost every type of cancer, with the highest rates in ovarian, esophageal, colorectal, head and neck, larynx, and lung cancers [26,105,106]. This shows that similar driver gene mutations may be found among different kind of tumors. Hence, two cancers with the same histologic origin may contain different molecular variants, affecting the overall evolution of the tumor and its response and resistance to targeted therapies. The high heterogeneity between different histotypes of lung cancer may provide an explanation for the great variation in treatment responses, as well as strategies that could be different for a single tumor or that are similar for different tumor types [13,19,20]. The most critical issues related to therapeutic strategies (sensitivity or resistance to the specific treatment) represent the heterogeneity between intra- and intertumors, primary tumor and metastasis, tumor cells and circulating tumor cells, and inter-single cells. Recent studies have provided insights into resistance mutations against osimertinib—i.e., a third-generation *EGFR* inhibitor that is designed to target T790M—and crizotinib, which is the *ALK* inhibitor [107,108].

There are various resistance mechanisms to osimertinib, such as the resistant mutation at the cysteine residue at position 797 (C797) that abolishes the direct covalent binding of osimertinib or resistance via the activation of bypass pathways, such as *MET* amplification [107,109]. In the case report by Ou et al. (2017) [110], the detection of an *EGFR* solvent front mutation (MAF of G796S/R) has been reported after treatment with osimertinib. Solvent front mutation, in addition to the C797 mutation, is potentially the dominant mutation driving resistance to osimertinib. In the case report by Chen et al. (2016) [111], not only the occurrence of the *EGFR* C797S mutation but also L792F/Y/H in three clinical subjects with acquired resistance to osimertinib treatment was observed. The study by Kim et al. (2013) [112] concerned heterogenous acquired resistance mechanisms in ALK-rearranged NSCLC treated with crizotinib. The study demonstrated secondary *ALK* mutations, a high *ALK* gene copy number, and the activation of *EGFR* signals expressed by a high *EGFR* polysomy and L858R in EGFR exon 21 after treatment with crizotinib.

The key step in lung cancer diagnosis and treatment can be the inclusion of molecular classification into the clinical classification of lung cancer. An example is a new nomenclature for the adenocarcinoma subtype proposed by The Cancer Genome Atlas Consortium (TCGA). The TCGA′s classification integrated the transcriptional subtypes with the histopathological, anatomic, and mutational categorizations and consists of three subtypes that are characterized by specific genetic alterations (Table 4) [56].

## 6. Summary

The early and accurate diagnosis of lung cancer is crucial for its effective treatment. Thanks to recent advancements in molecular strategies and analytic platforms, an increasing number of potential biomarkers in lung cancer diagnosis have been identified. Molecular biomarkers may be useful in diagnosis at an early and non-invasive stage of lung cancer in monitoring the course of the disease and the response to treatment, but none of them have been moved to the clinical setting yet. Data show that the genetic markers with the greatest diagnostic potential value in terms of lung cancer diagnosis are microRNAs and their expression profiles. Studies have reported that the false-positive rates of low-dose computer tomography (LDCT) could be reduced from 19.4% to 3.7% with an additional diagnosis conducted using a plasma miRNA panel. On the basis of the changes in the plasma microRNA, lung cancer could be predicted even 24 months earlier. This indicates that the microRNA panel has a potential to be used in order to detect lung cancer at early stages [5]. To exploit microRNA’s potential, it is necessary to standardize the quantification of circulating microRNAs and clarify the individual or environmental factors which affect the levels of circulating microRNA [83].

## Figures and Tables

**Figure 1 ijms-21-04569-f001:**
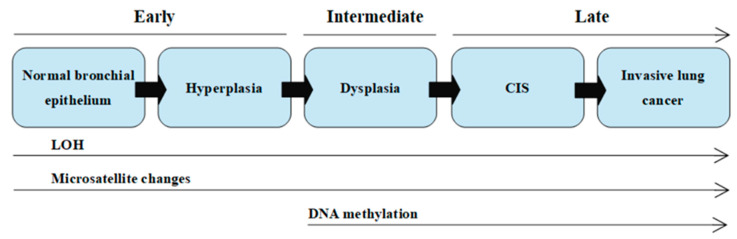
Scheme of the sequential changes during carcinogenesis in a simplified manner. LOH—loss of heterozygosity; CIS—carcinoma in situ. On the basis of (own modification of [7]).

**Table 2 ijms-21-04569-t002:** Techniques used frequently for mutation detection, based on [17].

Mutation Detection Techniques	Variant Types	
	SNVs	CNVs
Single-gene assays:		
Sanger sequencing	**+**	**-**
pyrosequencing	**+**	**-**
allele-specific PCR	**+**	**-**
single base extension	**+**	**-**
multiplex ligation-dependent probe amplification	**+**	copy number only
mass spectrometry	**+**	**-**
Gene-panel assays:		
amplicon-based panels	**-**	**+**
hybrid capture sequencing	**-**	**+**
next-generation sequencing	**+**	**+**
Fluorescence-based methods:		
fluorescence in situ hybridization	**-**	**+**
microarray-based CGH	**-**	**+**

Variant types are detected routinely (+) or cannot be detected (-). SNVs—single nucleotide variants, known as point mutations, small-scale deletions/insertions (indels); CNVs—copy number variants, including large-scale mutations such as amplifications, deletions, inversions, and translocations.

**Table 3 ijms-21-04569-t003:** Molecular landscape of non-small cell lung carcinoma (NSCLC) with the available treatment options, on the basis of [68].

Gene	Type of Genomic Aberrations	Frequency [%]	Currently Available Targeted Therapy *	Diagnostic Approaches	Ref.
Adenocarcinomas (ADC)					
*EGFR*	EGFR-TKI sensitizing mutations: EGFR exon 21, EGFR exon 19, G719X, L861Q point mutations Copy number variations (gains)	30–40	pemetrexed or bevacizumab therapy, afatinib, erlotinib, gefitinib, dacomitinib, osimertinib	PCR: sanger, real-time PCR, ddPCR, and NGS; IHC	[67,72,73,74,75,76,77]
*KRAS*	G12C mutation in *KRAS* gene	20–30	AMG-510	PCR, DNA sequencing	[67,72,73,74,77]
*MET*	MET exon 14 mutation (MET ex14), skipping mutations, overexpression, amplifications	2–5 3–4	skipping mutations—crizotinib, tepotinib; amplifications—crizotinib, capmatinib	mutations: sanger sequencing, NGS; amplifications: FISH, PCR, real-time PCR, NGS	[67,72,73,74,76,78]
*ALK*	ALK fusions	3–7	crizotinib, alectinib, ceritinib, brigatinib, lorlatinib	FISH (the gold standard); ALK-IHC has become a widely used technique with two validated antibodies in lung cancer (D5F3, 5A4)	[67,72,73,74,76,77,79]
*BRAF*	V600E mutation in *BRAF* gene; can co-exist with *KRAS* mutation	0.5–5	trametinib, dabrafenib	PCR: sanger, real-time PCR, and NGS	[67,72,73,74,76,78]
*ROS1*	ROS fusions	2–3	crizotinib	ROS1-IHC (screening) is still evolving (the use of the D4D6 rabbit monoclonal antibody) **; FISH; NGS	[67,72,73,74,76,77,80]
*RET*	RET rearrangements, gene fusion of KIF5B-RET; point mutations	1–2	vandetanib, cabozantinib, alectinib, BLU-667, LOXO-292	RT-PCR is typically combined with FISH; FISH; NGS	[67,72,73,74,76,78]
*NTRK*	NTRK rearrangements, gene fusions of NTRK1 (NTRKA), NTRK2 (NTRKB), NTRK3 (NTRKC)	1–2	entrectinib, larotrectinib, LOXO-195, repotrectinib	NGS with a panel that includes testing for NTRK1, NTRK2, NTRK3; IHC with subsequent confirmation by FISH or NGS	[67,72,73,74,76,78]
*HER2* ***	mutations in the kinase domain (exon 20), the most frequent is p.A775_G776insYVMA insertion amplifications, overexpressions	1–5 2–5	afatinib, dacomitinib, neratinib, trastuzumab, trastuzumab-emtansine, DS-8201a, poziotinib	mutations: PCR: sanger, real-time PCR and NGS; amplifications: FISH, NGS, real-time PCR	[72,74,76,78,81]
*PTEN* *PDGFRA* *PIK3CA* *TP53* *ERBB2* *TERT* *CDKN2A*	mutations copy number variations—gains losses	1.7 6–7 5 52 2–5 75 7	NA NA NA NA NA NA NA	- ****	[59,67,68]
Squamous cell carcinoma (SCC)					
*FGFR* *TP53* *NF1* *DDR2* *PDGFRA* *PIK3CA* *PTEN* *SOX2* *CDKN2A*	gene fusion of FGFR3-TACC3, mutations of FGFR1, FGFR2 tumor suppressor mutations, copy number variations (gains) mutations of NF1 point mutations of DDR2 amplification amplification tumor suppressor mutations, copy number variations (losses) amplification and copy number variation (gain) copy number variation (loss)	23 79 10 2–3 4 15 10 8 65 15	NA NA NA NA NA NA NA NA NA		[59,67,68]

* platinum-based chemotherapy (+/- pembrolizumab) is still the treatment of choice for patients without targetable mutations [82]; ** screening with ROS1-IHC and subsequent confirmation of IHC-positive cases with the use of FISH; ROS1-inhibitors should only be given to patients whose tumors are double positive according to IHC and FISH; *** *HER2* may be present in SCC but outside the kinase domain, with certain clinical benefit data when treating with afatinib; **** NGS can potentially test for all molecular alterations; NA—not available.

**Table 4 ijms-21-04569-t004:** The molecular classification of adenocarcinoma proposed by The Cancer Genome Atlas Consortium (TCGA) [97].

Name of the Unit	Abbreviation	Formerly	Mutations
Terminal respiratory unit	TRU	bronchioid	mutations in the *EGFR* gene and tumors expressing the kinase fusion;
Proximal-inflammatory	PI	squamoid	mutations in *NF1* and *TP53* genes
Proximal-proliferative	PP	magnoid	mutations of *KRAS* oncogene and inactivation of the *STK11* tumor suppressor gene

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
