# Peer review of "Genetic Markers in Lung Cancer Diagnosis: A Review"

_ijms, 2020, doi:10.3390/ijms21134569_

Round 1

Reviewer 1 Report

The authors present the review on genetic markers in lung cancer diagnostics. It is hot topic with wide consequences for patient treatment.

The authors do not discuss the clinical material available for this kind of diagnostics, i.e. cytological vs. histological material, its advantages and disadvantages and comparsion with liquid biopsies. The summary of available methodology would be also useful to list/table.

Figure 1: it is not the pathway through hyperplasia as the most important in pulmonary carcinogenesis, even hyperplasia switch to cancer is extremely rare in this localization. There is a pathway via squamous cell metaplasia to lead to squamous cell carcinoma.

Subchapter 4 should have own subchapters.

Author Response

We thank the Reviewer very much for valuable considerations. In the attachment is our full response to every comment and suggestion.

Reviewer 2 Report

The manuscript “GENETIC MARKERS IN LUNG CANCER DIAGNOSIS: A REVIEW” gives an extensive description of the main targets identified as biomarkers for lung cancer diagnosis. On the basis of the different genetic and epigenetic markers present in the tumor the authors indicate a different diagnostic approach.

Comments:

  • In the section “Genomics” the authors describe different strategies to identify lung cancer genetic markers. In this respect, they should report in the texts more examples of clinical and preclinical applications.
  • In the section “ Genomic and epigenomic changes in lung cancer diagnosis” the authors report in the Table 2 the different types of genomic aberrations and its frequency in the tumor. Where possible, they should report in the same table the diagnostic approach able to detect such aberrations.
  • In the “Abstract” the authors should better describe the aim of the manuscript
  • In the section “ Lung cancer genetic heterogeneity” the authors should report in the texts more recent references referred to clinical and preclinical studies reporting tumor heterogeneity as cause of resistance to therapies.

Author Response

(The authors gave the same response as above.)

Round 2

Reviewer 2 Report

The authors well addressed each comments in the previous review!